# Excess Heritability Contribution of Alcohol Consumption Variants in the “Missing Heritability” of Type 2 Diabetes Mellitus

**DOI:** 10.3390/ijms222212318

**Published:** 2021-11-15

**Authors:** Yujia Ma, Zechen Zhou, Xiaoyi Li, Zeyu Yan, Kexin Ding, Dafang Chen

**Affiliations:** Department of Epidemiology and Biostatistics, School of Public Health, Peking University, Beijing 100191, China; 2111110185@pku.edu.cn (Y.M.); peevesomega@163.com (Z.Z.); lvxi520@163.com (X.L.); 1510306132@pku.edu.cn (Z.Y.); 1610306225@pku.edu.cn (K.D.)

**Keywords:** type 2 diabetes, indirect genetic effects, heritability, behavior-related phenotypes

## Abstract

We aim to compare the relative heritability contributed by variants of behavior-related environmental phenotypes and elucidate the role of these factors in the conundrum of “missing heritability” of type 2 diabetes. Methods: We used Linkage-Disequilibrium Adjusted Kinships (LDAK) and LDAK-Thin models to calculate the relative heritability of each variant and compare the relative heritability for each phenotype. Biological analysis was carried out for the phenotype whose variants made a significant contribution. Potential hub genes were prioritized based on topological parameters of the protein-protein interaction network. We included 16 behavior-related phenotypes and 2607 valid variants. In the LDAK model, we found the variants of alcohol consumption and caffeine intake were identified as contributing higher relative heritability than that of the random variants. Compared with the relative expected heritability contributed by the variants associated with type 2 diabetes, the relative expected heritability contributed by the variants associated with these two phenotypes was higher. In the LDAK-Thin model, the relative heritability of variants of 11 phenotypes was statistically higher than random variants. Biological function analysis showed the same distributions among type 2 diabetes and alcohol consumption. We eventually screened out 31 hub genes interacting intensively, four of which were validated and showed the upregulated expression pattern in blood samples seen in type 2 diabetes cases. Conclusion: We found that alcohol consumption contributed higher relative heritability. Hub genes may influence the onset of type 2 diabetes by a mediating effect or a pleiotropic effect. Our results provide new insight to reveal the role of behavior-related factors in the conundrum of “missing heritability” of type 2 diabetes.

## 1. Introduction

Type 2 diabetes mellitus (type 2 diabetes) is a complex disease induced by a combination of environmental and genetic factors. Previous studies have shown that overweight, smoking, sedentary lifestyle and education are common risk factors of type 2 diabetes [1,2,3,4]. Meanwhile, genome-wide association studies (GWAS) have identified more than 500 susceptibility loci that demonstrated a robust association with type 2 diabetes [5]. In contrast to the tremendous stride in GWAS research, the conundrum of “missing heritability” in type 2 diabetes has progressed slowly and arduously. Genome-wide chip heritability analysis explained 19% of type 2 diabetes risk on a liability scale, which is much smaller when compared to heritability estimates expected from the observed trait concordance within families [6,7]. Although there are several hypotheses regarding rare variants, structural variants and gene–environment interactions for the missing heritability [8,9,10], the limited incremental value in heritability estimated by GWAS so far suggests that the genetic prediction of complex diseases on a population basis will be challenging. There is still a long way to go to fully understand the etiology of type 2 diabetes before getting it under control.

An important controversial assumption about heritability is the idea that the genetic influence on trait development can be separated from the environmental context [10]. In addition to the direct effect of genetics, part of the effect of genetic factors is mediated by environmental factors. Baud et al. found social genetic effects (SGE, effects of an individual’s genotypes on others’ phenotype, also called indirect genetic effects) can explain up to 29% of phenotypic variance, and for several traits, their contribution exceeded that of direct genetic effects (effects of an individual’s genotypes on its own phenotype) [11]. Undoubtedly, ignoring SGE can severely bias estimates of direct genetic effects (heritability) [11]. Xia et al. used a linear mixing model to estimate the indirect heritability between partners, and found evidence of indirect genetic effects between partners in about 50% of phenotypes [12]. The genetic nurturing effect proposed by Kong et al. is a manifestation of the social genetic effect within the family. Using results from a meta-analysis of educational attainment, they found the polygenic score computed for the non-transmitted alleles of 21,637 probands with at least one parent genotyped had an estimated effect on the educational attainment of the proband, that is 29.9% (*p* = 1.6 × 10^−14^) of that of the transmitted polygenic score [13]. The evidence above suggests that genetic factors can affect individual phenotypes through their contributions to the environment.

Another controversy about missing heritability is that there is currently much debate regarding the best model for how heritability varies across the genome. It has been shown that the LDAK model leads to estimates of common single-nucleotide polymorphism (SNP) heritability, on average, 43% (s.d. 3%) higher than those obtained from the widely used software Genome-wide Complex Trait Analysis (GCTA) and 25% (s.d. 2%) higher than those from the recently proposed extension LD and minor allele frequency (MAF) stratified multi-component GCTA (GCTA-LDMS) across 19 traits [14]. In terms of the rationality of the hypothesis, it is more realistic to employ the LDAK model, where expected heritability varies with both linkage disequilibrium (LD) and MAF [15,16]. In addition, considering the computational burden, the simplified LDAK-Thin model is also an alternative, which is a one-parameter model, and can be incorporated in any existing method simply by changing which predictors are included in the regression and how these are standardized [15].

In this study, we compared the heritability contribution of environmental phenotypes, especially behavior-related environmental phenotypes that have a genetic basis, with that of type 2 diabetes by using heritability estimation models to estimate the relative expected heritability tagged by each variant. The susceptibility variants of candidate environmental phenotypes were further characterized by functional annotation and protein–protein interaction (PPI) analysis to identify the potential key genes of type 2 diabetes. Our work is a new attempt to provide information and evidence to elucidate the genetic mechanisms underlying the missing heritability of type 2 diabetes and promote the development of comprehensive prevention for type 2 diabetes.

## 2. Results

### 2.1. Overview of Behavior-Related Phenotypes

Based on the results of the literature review and the results of Yuan et al., we eventually included 16 behavior-related phenotypes, including educational attainment, lifetime smoking index, alcohol consumption, coffee intake, caffeine intake, breakfast skipping, morningness, insomnia, sleep duration, short sleep, daytime napping, restless leg syndrome, moderate to vigorous physical activity, strenuous sports, vigorous physical activity and accelerometer. The union of variants for type 2 diabetes and the phenotype that both appear simultaneously in the tagging file is defined as the valid variant set for the consequent analysis. A total of 2607 valid variants were included in the analysis. The mean minimum allele frequency (MAF) was 0.28 (s.d. 0.14), and 149 variants were rare variants (MAF < 0.05). The results of traditional epidemiological studies on behavior-related phenotypes of type 2 diabetes and the information of susceptibility variants for each phenotype included in the analysis are shown in Table 1 and Table 2, and Figure 1.

### 2.2. Estimation of Relative Expected Heritability by LDAK

The relative expected heritability estimated of all 2607 variants estimated by SumHer under the LDAK model assumption was 19.5, which was not higher than that of simulated sampling. All variants of behavior-related phenotypes accounted for 83.39% of the total phenotypic heritability. Educational attainment contributed the most, at 76.43% of the total phenotypic heritability. The heritability contributed by the susceptibility variants was significantly correlated with the number of variants (correlation coefficient = 0.90, *p* < 0.001), as seen in Table 3. The results of simulation sampling showed that the relative heritability of the susceptibility variants of caffeine intake and alcohol consumption was significantly higher than that of random variants. In caffeine intake, the average heritability of the total variants was 0.01 and the average heritability of phenotypic variants was 0.04, while the attribution heritability of phenotypic variants was 2.43% and the relative heritability of phenotypic variants was 4.51 times. The corresponding parameters for alcohol consumption were 0.01, 0.02, 37.45% and 2.24 times, respectively. The relative heritability of phenotypic variants of skipping breakfast, coffee consumption and strenuous sports were also more than 2 times that of type 2 diabetes variants, while it was not statistically significant compared with simulation sampling.

### 2.3. Estimation of Relative Expected Heritability by LDAK-Thin

The relative expected heritability estimated of all 2607 variants estimated by SumHer under the LDAK-Thin model assumption was 671.3, which was significantly higher than that of simulated sampling. All variants of behavior-related phenotypes accounted for 86.88% of total phenotypic heritability. Educational attainment contributed the most, at 79.48% of the total phenotypic heritability. The heritability contributed by the susceptibility variants was significantly correlated with the number of variants (correlation coefficient = 0.91, *p* < 0.001), as seen in Table 4. Compared to the simulation sampling, the relative heritability of variants of 11 phenotypes, including insomnia, educational attainment, lifetime smoking index, alcohol consumption, coffee consumption, daytime napping, sleep duration, short sleep, morningness, moderate to vigorous physical activity and vigorous physical activity, was statistically higher than the random variants. Among the phenotypes with significant differences, the relative heritability of phenotypic variants of all phenotypes was higher than that of type 2 diabetes, except alcohol consumption. The relative heritability of phenotypic variants for short sleep was the highest, which was 1.36 times that of type 2 diabetes, which accounted for 8.87% of the total phenotypic heritability.

### 2.4. Biological Function Analysis

As the variants of alcohol consumption were identified to contribute higher relative heritability than that of the random variants in two heritability models, we then performed the functional annotation, enrichment analysis and protein interaction network analysis for the targeted phenotypes (Appendix A).

For 98 susceptibility variants associated with alcohol consumption, 7 of them (7.14%) were missense mutations, 1 (1.02%) was a synonymous mutation (rs17029090), 14 (14.29%) were in the untranslated region (UTR) and 66 (67.35%) were mutations in the intronic region. The regulatory element functional annotation results revealed six transcription factor binding sites and two CpG sites. Fourteen variants had Combined Annotation–Dependent Depletion (CADD) scores greater than 12.37, suggesting that they might be deleterious mutations (Appendix A). The Chi-Squared test showed no significant difference between the distributions for the functional category and RegulomeDB ranking among type 2 diabetes and alcohol consumption (Appendix A).

KEGG pathway enrichment analysis was performed on 55 genes annotated by susceptibility variants of alcohol consumption (Appendix A). Under the false discovery rate at the 0.05 level, the study found that the genes were significantly enriched in glycolysis/gluconeogenesis (hsa00010), tyrosine metabolism (hsa00350), fatty acid degradation (hsa00071), retinol metabolism (hsa00830), metabolism of xenobiotics by cytochrome P450 (hsa00980), drug metabolism-cytochrome P450 (hsa00982), chemical carcinogenesis (hsa05204) and propanoate metabolism (hsa00640).

### 2.5. Screening of Hub Genes

Based on the closeness, edge percolated component (EPC), and maximum neighborhood component (MNC) stress of the protein interaction network formed by genes annotated from variants associated with alcohol consumption and type 2 diabetes, the genes in the protein interaction network were sorted (Appendix A). We eventually screened out 31 hub genes interacting intensively (*p* < 0.001), as seen in Figure 2, of which 2 genes (*GCKR* and *TCF4*) were identified simultaneously by the susceptibility variants of alcohol consumption and type 2 diabetes (Appendix A). 

*GCKR* is highly expressed in the liver. *CAMD2* and *RPTOR* were identified by the susceptibility variants of alcohol consumption only, of which *CAMD2* is highly expressed in brain-related tissues (Appendix A).

### 2.6. Significant Upregulation of RPTOR

Among the hub genes screened, four genes (*NEUROG3*, *TCF7L2*, *MAP2K5* and *RPTOR*) were validated as showing the upregulated expression pattern in blood samples in type 2 diabetes cases (Figure 3).

## 3. Discussion

This study provides new insight into the association between type 2 diabetes and alcohol consumption. In this study, we found the variants of alcohol consumption were identified as contributing higher relative heritability than that of the random variants in two heritability models. In the LDAK model, the relative expected heritability contributed by the variants associated with these two phenotypes was twice as much as the relative expected heritability contributed by the variants associated with type 2 diabetes, while in the LDAK-Thin model, the relative expected heritability contributed by the variants associated with these two phenotypes was less than the relative expected heritability contributed by the variants associated with type 2 diabetes. Such inconsistencies in the relative expected heritability of each variant may be due to differences in the weights assigned to the variants in model assumptions. Boyle et al. believed that the heritability of a typical complex phenotype is driven by a large number of variations in the regulatory element region [17]. Liu et al. found that the distribution of heritability in each variant showed tissue specificity, with genes with related functions (e.g., neuronal function in schizophrenia and immune function in Crohn’s disease) contributing slightly more to heritability than random genes, while genes not expressed in related cell types did not contribute to heritability [18]. Yet, the specific assumptions of which model is more reasonable still need to be further explored at the level of biological mechanisms. Biological function analysis showed the same distributions for the functional category and RegulomeDB ranking among type 2 diabetes and alcohol consumption. Based on the topological parameters of the protein interaction network, we eventually prioritize an intensively interactive hub 31 genes, of which two hub genes (*CAMD2* and *RPTOR*) were annotated by the variants of alcohol consumption only. Our study provided a comprehensive approach to delineate the potential causal genes and biological processes involved in type 2 diabetes pathogenesis and proposed new insight into revealing the role of behavior-related environmental factors in the conundrum of “missing heritability” of type 2 diabetes.

Systematic reviews have found a U-shaped association between alcohol consumption and type 2 diabetes [19,20]. Moderate alcohol consumption also has a protective effect on blood glucose management. Initiating moderate wine intake, especially red wine, among well-controlled diabetics as part of a healthy diet is apparently safe and modestly decreases cardiometabolic risk. In particular, only alcohol dehydrogenase allele [*ADH1B**1] carriers significantly benefited from the effect of both wines on glycemic control compared with persons homozygous for *ADH1B**2 [21]. We found that the *ADH1B* gene is a missense mutation annotated by the variant rs1229984 associated with alcohol consumption, which implied that it may be a key gene in the biological mechanism of alcohol consumption and type 2 diabetes. However, this gene was not tagged as a hub gene in our study, possibly because the number of genes annotated by variants of type 2 diabetes exceeded that of alcohol consumption, thus it may be diluted by type 2 diabetes-related genes.

Among the hub genes identified, we particularly highlighted those annotated by alcohol consumption variants, because these genes may influence the onset of type 2 diabetes by a mediating effect or a pleiotropic effect, which is of significance for the comprehensive prevention of type 2 diabetes. *GCKR*, a hub gene identified simultaneously by the susceptibility variants of alcohol consumption and type 2 diabetes, has densely interacted with type 2 diabetes-related genes such as *FTO* and *SLC2A2*. GCKR is the susceptibility gene candidate of maturity-onset diabetes of the young (MODY), whose protein product binds non-covalently to form an inactive complex with the enzyme to regulate glucokinase in liver and pancreatic islet cells. Previous studies have found that polymorphisms in *GCKR* (rs780094) are associated with non-alcoholic fatty liver disease in multiple populations [22,23,24]. Evidence of an association between this variant and type 2 diabetes or metabolic risk has also been detected [25,26]. An exome-chip association analysis for circulating FGF21 levels in Chinese individuals found that the common missense variant of *GCKR*, rs1260326 (p.Pro446Leu), may influence *FGF21* expression via its ability to increase glucokinase (GCK) activity [27]. This can lead to enhanced *FGF21* expression via elevated fatty acid synthesis, which is recognized as an important metabolic regulator of glucose homeostasis [27,28]. *CAMD2* and *RPTOR* were specifically alcohol consumption annotating genes. *CADM2* variants influence a wide range of both psychological and metabolic traits, suggesting common biological mechanisms across phenotypes via the regulation of *CADM2* expression levels in adipose tissue [29]. *RPTOR* encodes a component of a signaling pathway that regulates cell growth in response to nutrient and insulin levels. Its encoded protein forms a stoichiometric complex with the mTOR kinase, of which the dysregulation of signaling is implicated in pathologies that include diabetes, cancer and neurodegeneration [30]. Regarding the indirect effect of genetic factors, our study calculated the heritability contribution of each phenotype and explored the biological function of the potential mechanism. Such a new method identified genes related to the onset of type 2 diabetes, and the function of these pleiotropic genes needs to be verified in subsequent analyses using primary individual-level data or experimental evidence.

There are some limitations in this study. Firstly, due to the limitation of computational resources, only two simple heritability models were considered, and the models weighted by functional annotation were ignored. Since the estimated heritability in this study is the relative expected heritability rather than the absolute heritability, the results between models were not comparable to a certain extent. Although we applied the relative heritability of phenotypic variants, the results of some phenotypes were not consistent. The hypothesis relating to which model is more reasonable still needs to be further explored. In particular, whether this phenomenon exists in more complex heritability models also needs to be followed up. In addition, the extrapolation of the conclusions in non-European ancestry needs to be further verified as there are systematic differences not only in gene frequency among different populations, but also in their behavior and lifestyle, such as drinking culture. Further studies on a larger scale are needed to verify the reliability of the conclusions in other populations.

Previous studies identified hub genes of type 2 diabetes based on the direct genetic effect, while recent studies found that the majority of phenotypic variance is driven by genes that are not directly related to the phenotypes [18]. Therefore, indirect effects of genetic factors, especially those mediated by modifiable phenotypes such as behavior-related phenotypes, should be considered in etiological studies and intervention strategies for chronic diseases such as type 2 diabetes.

## 4. Materials and Methods

### 4.1. Identification for Candidate Environmental Phenotypes Associated with Type 2 Diabetes

Behavior-related environmental phenotypes found to be potentially causally associated with type 2 diabetes were identified as candidate phenotypes based on previous traditional epidemiological literature reports and Mendelian randomization studies. The literature was searched in the PubMed database, and the search strategies were as follows: ((((((((((meta-analysis [Publication Type]) OR meta-analysis [Title/Abstract]) OR meta-analysis [Title/Abstract]) OR meta-analysis [Title/Abstract]) OR meta-analysis [Title/Abstract]) OR SystematicReview [Publication Type]) OR systematic review [Title/Abstract])) AND ((Risk Factors [MeSH Terms]) OR risk factor [Title/Abstract])) AND (((Diabetes Mellitus, Type 2 [MeSH Terms]) OR Type 2 diabetes [Title/Abstract]) OR Type 2 diabetes mellitus [Title/Abstract]))“. In addition, we also refer to the wide-angled Mendelian randomization study of Yuan et al. [31]. Phenotypes in the categories of “lifestyle and sleep-related factors” and “education” were selected, and the phenotypes whose variants were from European ancestry were recorded as candidate phenotypes.

### 4.2. The Data Source

Genetic variants information of type 2 diabetes was acquired from Mahajan et al. ’s work [32]. In this study, GWAS results from 32 studies for 898,130 individuals (74,124 T2D cases and 824,006 controls) of European ancestry were aggregated. Imputation was implemented using the Haplotype Reference Consortium reference panel. Association summary statistics from sex-combined analyses for each variant across all studies were aggregated using fixed-effect meta-analyses with an inverse-variance weighting of log-ORs and corrected for residual inflation by means of genomic control. In total, 403 independent association signals were detected by conditional analyses at each of the genome-wide-significant risk loci for type 2 diabetes (except at the major histocompatibility complex (MHC) region). Summary-level data are available at the DIAGRAM consortium (http://diagram-consortium.org/, accessed on 13 November 2020) and Accelerating Medicines Partnership type 2 diabetes (http://www.type2diabetesgenetics.org/, accessed on 13 November 2020). The information of susceptibility variants of candidate phenotypes is shown in Table 1. Detailed definitions of each phenotype are shown in Appendix A.

### 4.3. LDAK Model

The LDAK model [14] is an improved model to overcome the equity-weighted defects for GCTA, which weighted the variants based on the relationships between the expected heritability of an SNP and minor allele frequency (MAF), levels of linkage disequilibrium (LD) with other SNPs and genotype certainty. When estimating heritability, the LDAK Model assumes:(1)$$E[h_{j}^{2}]\propto {\left[f_{i}(1-f_{i})\right]}^{1+\alpha }\times \varpi_{j}\times r_{j}$$
where $E[h_{j}^{2}]$ is the expected heritability contribution of SNP_j_ and *f_j_* is its (observed) MAF. The parameter α determines the assumed relationship between heritability and MAF. In human genetics, it is commonly assumed that heritability does not depend on MAF, which is achieved by setting α = –1; however, we consider alternative relationships. The SNP weights $\varpi_{1}$,……, $\varpi_{m}$ are computed based on local levels of LD; $\varpi_{j}$ tends to be higher for SNPs in regions of low LD, and thus the LDAK Model assumes that these SNPs contribute more than those in high-LD regions. Finally, $r_{j}$ ∈ [0,1] is an information score measuring genotype certainty; the LDAK Model expects that higher-quality SNPs contribute more than lower-quality ones.

### 4.4. LDAK-Thin Model

The LDAK-Thin model [15] is a simplification of the LDAK model. The model assumes $\varpi_{j}$ is either 0 or 1, that is, not all variants contribute to the heritability based on the LDAK model.

### 4.5. Model Implementation

We applied SumHer (http://dougspeed.com/sumher/, accessed on 13 January 2021) [33] to estimate each variant’s expected heritability contribution. The reference panel used to calculate the tagging file was derived from the genotypes of 404 non-Finnish Europeans provided by the 1000 Genome Project. Considering the small sample size, only autosomal variants with MAF ≥ 0.01 were considered. Data preprocessing was completed with PLINK1.9 (https://www.cog-genomics.org/plink/1.9/, accessed on 13 January 2021) [34]. SumHer analysies are completed using the default parameters, and a detailed code can be found in http://dougspeed.com/reference-panel/, accessed on 13 January 2021.

### 4.6. Estimation and Comparison of Expected Heritability

To estimate and compare the relative expected heritability, we define three variants set in the tagging file: *G*_1_ was generated as the set of significant susceptibility variants for type 2 diabetes; *G*_2_ was generated as the union of type 2 diabetes and the set of each behavior-related phenotypic susceptibility variants. Simulation sampling is conducted because all estimations calculated from tagging file were point estimated without a confidence interval. We hoped to build a null distribution of the heritability of random variants. This allowed us to distinguish the contribution of phenotypic variants from the null distribution of random variants at the significance level of α = 0.05. Therefore, the random variant set *G*_3_ was generated. *G*_3_ was defined as the union of type 2 diabetes and the random susceptibility variants. The set size of *G*_3_ was equal to that of *G*_2_, which could control spurious inflation caused by increasing the number of variants. The calculation procedure for *G*_3_ is the exactly same as that for *G*_2_. The sum of expected relative heritability contributed to by variants in *G*_1_ and *G*_2_ was calculated, respectively. We simulated random sampling progress to generate 100 *G*_3_ sets to the significance of *G*_2_ at the level of α = 0.05. For each phenotype, we also calculated indexes as follows:

Average heritability of total variants
(2)$${\bar{h}}_{total}^{2}=\frac{h^{2}{}_{G_{2}}}{n_{G_{2}}}$$

Average heritability of phenotypic variants
(3)$${\bar{h}}_{pheno}^{2}=\frac{h^{2}{}_{G_{2}}-h^{2}{}_{G_{1}}}{n_{G_{2}}-n_{G_{1}}}$$

Attribution heritability of phenotypic variants
(4)$$AHPV=\frac{h^{2}{}_{G_{2}}-h^{2}{}_{G_{1}}}{h^{2}{}_{G_{2}}}\times 100\%$$

Relative heritability of phenotypic variants (5)$$RHPV=\frac{\frac{h^{2}{}_{G_{2}}-h^{2}{}_{G_{1}}}{n_{G_{2}}-n_{G_{1}}}}{\frac{h^{2}{}_{G_{1}}}{n_{G_{1}}}}$$

$h^{2}{}_{G_{i}}$ and $n_{G_{i}}(i=1,2,3)$ were the expected relative heritability and set size of *G*_1_, *G*_2_ and *G*_3_.

### 4.7. Biological Function Analysis

#### 4.7.1. Functional Annotation

Susceptibility variants were annotated by SNPNexus (https://www.snp-nexus.org/v4/, accessed on 8 February 2021) [35]. SNPNexus is a web-based annotation tool developed by Claude Chelala et.al. The latest version was updated in December 2019. CADD scores greater than 12.37 were considered high probability of a harmful mutation [36]. Potential regulatory functions were annotated by RegulumeDB [37]. We also explored the expression of hub genes in the dataset Genotype-Tissue Expression (GTEx) [38] using FUMA [39].

#### 4.7.2. KEGG Pathway Enrichment Analysis

To clarify the biological mechanism behind the potential pathogenic genes of type 2 diabetes behavior-related phenotypes, we conducted pathway enrichment analysis on the susceptibility variants of type 2 diabetes in Kyoto Encyclopedia of Genes and Genomes (KEGG) dataset [40] behavior-related phenotypes annotated by GRCH37/HG19. An over-represented analysis was used to test whether potential pathogenic genes of behavior-related phenotypes of type 2 diabetes were significantly enriched in the above pathways. The data targeted by over-representative analysis is a group of genes of interest. The statistical principle is the hypergeometric distribution test, and the *p*-value is calculated by Fisher’s exact probability method [41]. The *p*-value in the target pathway (KI) is calculated as follows:(6)$$P(K_{i})=1-\sum \frac{\left(\frac{M}{n}\right)\left(\frac{N-M}{n-m}\right)}{\left(\frac{N}{n}\right)}$$

Among them, *N* is the total number of genes studied, *N* is the total number of potential pathogenic genes for behavior-related phenotypes of type 2 diabetes, *M* is the total number of genes in pathway $K_{i}$ and *M* is the total number of potential pathogenic genes for behavior-related phenotypes of type 2 diabetes in pathway $K_{i}$. Subsequently, the Benjamini and Hochberg method was used to correct the multiple tests, and the significance level of pathway analysis was defined as the false discovery rate (FDR) < 0.05.

#### 4.7.3. Protein Interaction Network Analysis

Based on the “guilt-by-association” principle, protein–protein interaction (PPI) analysis identifies a set of genes whose downstream products (proteins) are associated with each other. These identified genes combine to influence disease. In this study, protein interaction network analysis was completed by String (https://string-db.org/, accessed on 8 February 2021) [42]. String is a database of known and predicted protein–protein interactions designed to collect, score and integrate all publicly available sources of protein–protein interaction information, and to supplement the information by calculating the predictions. The visualization of the network was completed by Cytoscape 3.7.0 [43].

#### 4.7.4. Screening of Hub Genes

The acquisition of hub proteins and subnetworks in the complex differential protein interaction network is particularly important for the search of the mechanism of life activities. Therefore, in this study, the Cytohubba module [44] was used to sequence the genes in the network and screen the hub genes. Cytohubba can predict and explore key nodes and subnetworks in a given network through several topological algorithms. We used four global topology analysis methods, including the Edge Percolated Component (EPC), Maximum Neighborhood Component (MNC) and centralities based on shortest paths including closeness and stress to prioritize genes in the network. Hub genes were defined as the shared top 25% of genes sorted by each method.

#### 4.7.5. Expression Analysis of Hub Gene in Blood Samples

To evaluate whether the hub genes identified are differentially expressed, we used publicly available expression dataset GSE184050 from the Gene Expression Omnibus (https://www.ncbi.nlm.nih.gov/geo/, accessed on 29 September 2021) database. GSE184050 compared changes in gene expression using two longitudinally collected blood samples from subjects who transitioned to type 2 diabetes between the time points against those who did not with a novel analytical network approach. A total of 116 individual samples (50 from type 2 diabetes cases and 66 from healthy controls) were submitted to the analysis. RNA was extracted, amplified, reverse transcribed, labelled and sequenced with Illumina HiSeq 2000 (Illumina, Inc., San Diego, CA, USA).

## 5. Conclusions

We found that alcohol consumption contributed higher relative heritability and eventually screened out 31 hub genes candidate of the development of type 2 diabetes. Hub genes may influence the onset of type 2 diabetes by a mediating effect or a pleiotropic effect. Our results provided new insight into revealing the role of behavior-related factors in the conundrum of the “missing heritability” of type 2 diabetes.

## Figures and Tables

**Figure 1 ijms-22-12318-f001:**
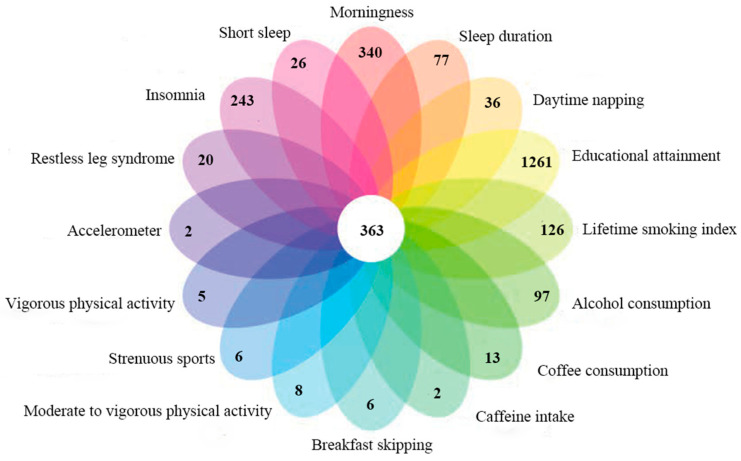
An overview of behavioral related phenotypes’ susceptibility variants for type 2 diabetes mellitus.

**Figure 2 ijms-22-12318-f002:**
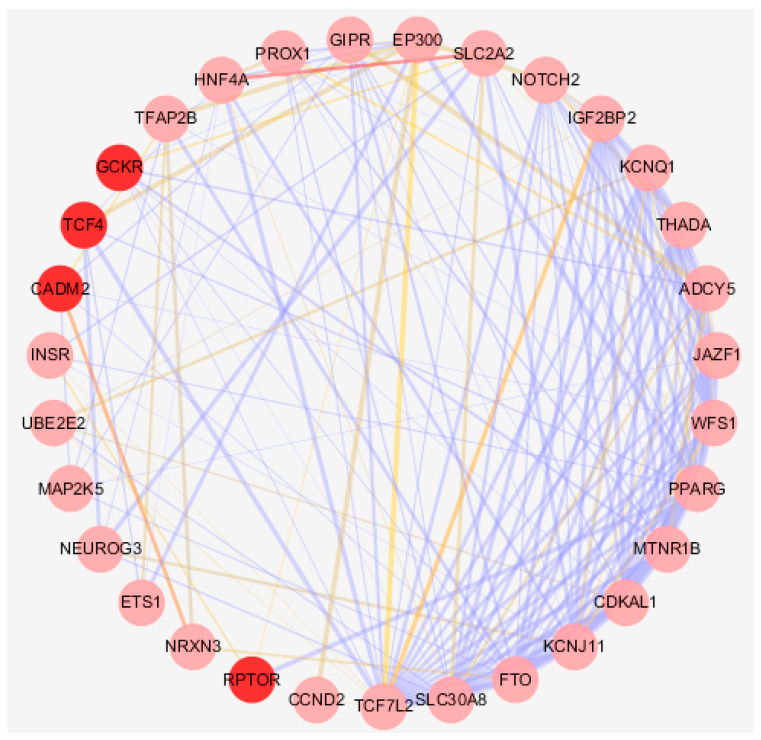
Protein–protein interaction (PPI) of hub genes prioritized by topological parameters. The deepened red nodes are genes annotated by variants of alcohol consumption. The thickness of the edges indicates the strength of data support. The color of the edges indicates the intensity of co-expression between two nodes; the brighter the color is, the higher the co-expression intensity is.

**Figure 3 ijms-22-12318-f003:**
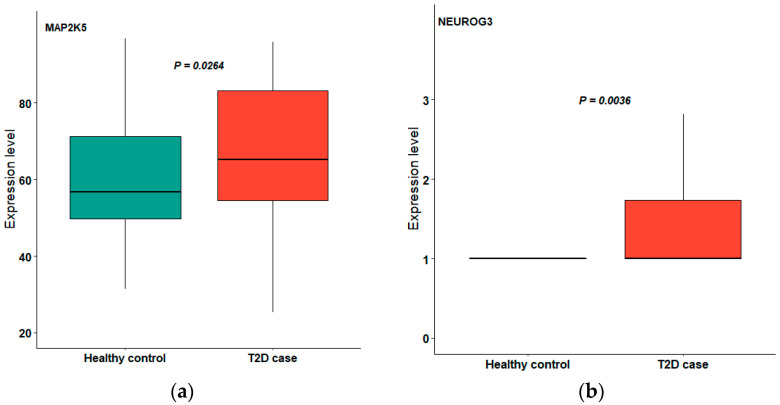
Upregulated expression of *MAP2K5* (**a**), *NEUROG3* (**b**), *RPTOR* (**c**) and *TCF7L2* (**d**) in periodontitis.

**Table 1 ijms-22-12318-t001:** Information on type 2 diabetes related behavioral phenotypic susceptibility variants.

Factors	PMID	Year	Case	Control	Unit
Alcohol consumption	30643251	2019	941,280	NA	Drinks/week
Coffee consumption	31046077	2019	375,833	NA	NA
Caffeine intake	21490707	2011	47,341	NA	mg/d
Breakfast skipping	31190057	2019	193,860	NA	NA
Lifetime smoking index	31689377	2019	462,690	NA	SD
Daytime napping	31409809	2019	452,071	NA	Events
Sleep duration	30846698	2019	446,118	NA	Hours/d
Short sleep	30846698	2019	106,192	305,742	Events
Long sleep	30846698	2019	34,184	305,742	Events
Insomnia	30804565	2019	397,972	933,038	Events
Morningness	30696823	2019	372,765	278,530	Events
Restless leg syndrome	29029846	2017	15,126	95,725	Events
Moderate to vigorous physical activity	29899525	2018	377,234	NA	SD
Strenuous sports	29899525	2018	124,842	225,650	≥2–3 vs. 0 day/weeks
Vigorous physical	29899525	2018	98,060	162,995	≥3 vs. 0 day/weeks
Accelerometer	29899525	2018	91,084	NA	NA
Educational attainment	30038396	2018	1,131,881	NA	SD

NA, missing value; SD, standard deviation.

**Table 2 ijms-22-12318-t002:** Distribution of susceptibility variants for behavior-related phenotypes in type 2 diabetes.

Behavior-Related Phenotypes	Variants Reported in Literature	Variants in Tagging File	Valid Variants Analyzed	MAF (s.d.)	EAF (s.d.)
Type 2 diabetes	403	363	363	0.25(0.14)	0.48(0.28)
Educational attainment	1272	1263	1624	0.27(0.14)	0.46(0.26)
Lifetime smoking index	126	126	489	0.31(0.12)	0.49(0.22)
Alcohol consumption	99	98	460	0.27(0.14)	0.44(0.27)
Coffee consumption	15	14	376	0.25(0.14)	0.39(0.26)
Breakfast skipping	6	6	369	0.21(0.08)	0.50(0.31)
Caffeine intake	2	2	365	0.34(0.07)	0.34(0.04)
Morningness	351	342	703	0.30(0.13)	0.46(0.23)
Insomnia	248	244	606	0.30(0.13)	0.49(0.24)
Sleep duration	78	77	440	0.29(0.13)	0.52(0.25)
Daytime napping	37	36	399	0.34(0.12)	0.57(0.20)
Short sleep	27	26	389	0.29(0.11)	0.50(0.25)
Restless leg syndrome	20	20	383	0.28(0.13)	0.51(0.27)
Moderate to vigorous physical activity	9	9	371	0.26(0.16)	0.47(0.29)
Strenuous sports	6	6	369	0.26(0.16)	0.53(0.28)
Vigorous physical	5	5	368	0.34(0.14)	0.34(0.14)
Accelerometer	2	2	365	0.29(0.03)	0.72(0.10)
Total	2674	2607	2607	0.28(0.14)	0.47(0.25)

MAF, Minor Allele Frequency; EAF, Effect Allele Frequency.

**Table 3 ijms-22-12318-t003:** Estimated of relative expected heritability by LDAK.

Phenotypes	Expected Heritability					
	Estimation	Simulation (s.d.)	${\bar{h}}_{total}^{2}$	${\bar{h}}_{pheno}^{2}$	AHPV	RHPV
Type 2 diabetes	3.2	-	0.01	-	-	-
Caffeine intake *	3.3	3.3(0.0)	0.01	0.04	2.43	4.51
Alcohol consumption *	5.2	4.3(0.3)	0.01	0.02	37.45	2.24
Breakfast skipping	3.4	3.3(0.1)	0.01	0.02	4.06	2.56
Coffee consumption	3.5	3.4(0.1)	0.01	0.02	7.73	2.34
Strenuous sports	3.3	3.3(0.1)	0.01	0.02	3.26	2.04
Moderate to vigorous physical activity	3.3	3.3(0.1)	0.01	0.01	2.33	1.08
Educational attainment	13.7	16.2(1.2)	0.01	0.01	76.43	0.93
Insomnia	5.2	5.7(0.4)	0.01	0.01	37.45	0.89
Morningness	4.8	6.6(0.6)	0.01	0.00	32.01	0.50
Lifetime smoking index	3.7	4.5(0.4)	0.01	0.00	13.42	0.45
Short sleep	3.3	3.5(0.1)	0.01	0.00	2.53	0.36
Sleep duration	3.4	4.1(0.3)	0.01	0.00	0.30	0.22
Vigorous physical	3.2	3.3(0.1)	0.01	0.00	4.53	0.22
Restless leg syndrome	3.3	3.4(0.1)	0.01	0.00	0.88	0.16
Daytime napping	3.2	3.6(0.2)	0.01	0.00	0.00	0.00
Accelerometer	3.2	3.3(0.1)	0.01	0.00	0.00	0.00
Total	19.5	26.7(1.6)	0.0075	0.0072	83.39	0.81

* indicated that the estimation was significantly higher than the simulation sampling results at the significance level of α = 0.05. AHPV, Attribution Heritability of Phenotypic Variants; RHPV, Relative heritability of phenotypic variants.

**Table 4 ijms-22-12318-t004:** Estimation of relative expected heritability by LDAK-Thin.

Phenotypes	Expected Heritability					
	Estimation	Simulation (s.d.)	${\bar{h}}_{total}^{2}$	${\bar{h}}_{pheno}^{2}$	AHPV	RHPV
type 2 diabetes	88.1	-	0.24	-	-	-
Short sleep *	96.7	91.4(1.1)	0.25	0.33	8.87	1.36
Daytime napping *	99.7	92.5(1.1)	0.25	0.32	11.64	1.33
Strenuous sports	89.9	88.9(0.5)	0.24	0.31	2.05	1.27
Coffee consumption *	91.7	89.3(0.7)	0.24	0.28	3.90	1.13
Educational attainment *	429.3	245.9(8.0)	0.26	0.27	79.48	1.11
Insomnia *	151.2	118.4(3.4)	0.25	0.26	41.73	1.07
Caffeine intake	88.6	88.4(0.3)	0.24	0.26	0.58	1.06
Sleep duration *	107.1	97.9(1.8)	0.24	0.25	17.73	1.02
Lifetime smoking index *	118.9	104.1(1.8)	0.24	0.24	25.91	1.01
Morningness *	171.7	130.2(3.4)	0.24	0.25	48.68	1.01
Moderate to vigorous physical activity *	90.1	88.8(0.6)	0.24	0.24	2.17	1.01
Alcohol consumption *	109.7	99.9(1.9)	0.24	0.22	19.70	0.92
Breakfast skipping	89.1	88.8(0.4)	0.24	0.16	1.07	0.65
Vigorous physical	89.0	88.8(0.4)	0.24	0.18	1.01	0.74
Restless leg syndrome	90.4	92.7(0.9)	0.24	0.12	2.59	0.48
Accelerometer	88.1	88.3(0.3)	0.24	0.00	0.00	0.00
Total	671.3	324.4(9.0)	0.2575	0.2599	86.88	1.07

* indicated that the estimation was significantly higher than the simulation sampling results at the significance level of α = 0.05. AHPV, Attribution Heritability of Phenotypic Variants; RHPV, Relative heritability of phenotypic variants.

## Data Availability

Our datasets analyzed during the current study were derived from the following public domain resources: Summary statistics of the GWAS is available from DIAGRAM consortium (http://diagram-consortium.org/, accessed on 13 November 2020). We applied SumHer (http://dougspeed.com/sumher/, accessed on 13 January 2021) to estimate each variant expected heritability contribution. The reference panel used to calculate the tagging file was derived from the genotypes of 404 non-Finnish Europeans provided by the 1000 Genome Project. Data preprocessing was completed with PLINK1.9 (https://www.cog-genomics.org/plink/1.9/, accessed on 13 January 2021).

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
