# Peer review of "Excess Heritability Contribution of Alcohol Consumption Variants in the “Missing Heritability” of Type 2 Diabetes Mellitus"

_ijms, 2021, doi:10.3390/ijms222212318_

Round 1
Reviewer 1 Report
- Please include a conclusion and a graphical abstract or summary.
- Define all abbreviations the first time they appear in abstract and main text.
- In line 88-87, need more detailed explanation for "0.28(0.14)".
- In table 1 , confirm whether the "peer'' is correct
- In table 2, need explanation for "EAF"/"0.25(0.14)"/"0.48(0.28)"
- In figure1, the distribution seems like all same, but not same,please make it accordingly.
- In table 3, need explanation for "3.3(0.0)"/AHPV/RHPV.
- Line 117,is it “a”?

Author Response
I am fully grateful for your patience and the valuable suggestions you've given us. The response to your suggestion is detailed in the Word document. Please see the attachment.
I am looking forward to having further communication with you. Thanks again most warmly.

Reviewer 2 Report
The aim of this paper is the exploration of “missing heritability” of type 2 diabetes focusing on alcohol consumption variants. The paper is well structured and well written and the hypothesis is clear. Some comments and advices:
- In the abstract (line 14) and in the main text please define what “valid variants” means? How did you choose 2607 valid variants? Can you define valid? There are some inclusion criteria or cut-off?
- How did you choose 16 behavior-related phenotypes? What is your criteria? How do adjust for correlation between traits? For example, I suppose that coffee and caffeine intake or sleep duration and short sleep are highly correlated.
- It is not clear what simulation sampling is? Please, explain more the setting to simulate random sampling progress. Why do you choose n=100 G3 sets?
- Please provide more details of this population (baseline characteristics, sample sizes, n. of patients with type of 2 diabetes).
- Do you consider to adjust the model to estimate H2 of type 2 diabetes with the other variables? or for PCA? How many principal component?
- Define in legend AHPV and RHPV in Table 3
- “The relative heritability contributed both phenotypes’ variants was over 2 times of that of type 2 diabetes.” Please, specify this interpretation. Again, please, explain better lines 186-188 (the relative expected heritability of each variant of alcohol consumption is 2.24 times with respect to that of type 2 diabetes, while this parameter is estimated less than one in the LDAK-Thin model). The interpretation of this result is: type of 2 diabetes has a genetic component and the remains is due to alcohol assumption? Is this the “missing heritability”?
- Again, how can you interpret the relative expected heritability? Please, explain for example relative expected heritability estimated was 671.3.
- To have a global “missing heritability” for type 2 diabetes, I suggest to estimate it considering all 16 behavior-related phenotypes together not independently
- Why the enrichment analysis is performed only for alcohol assumption? What about the other traits?
- Figure 3 (b) – How many samples are in the healthy control NEUROG3? P-value has no sense if the samples size are inadeguate.
- Can you explain better the association between the estimation of 16 behavior-related phenotypes and “missing heritability”. If your aim is to estimate “missing heritability” of type 2 diabetes, authors can explore directly the heritability of type 2 diabetes but select the low minor allele frequency (MAF) and explore these variants.
Author Response
I am fully grateful for your patience and the valuable suggestions you've given us. The response to your suggestion is highlighted in the Word document. Please see the attachment.
I am looking forward to having further communication with you. Thanks again most warmly.
